# Efficacy and mechanism of action of cipargamin as an antibabesial drug candidate

Hang Li[1†], Shengwei Ji[1,2†], Nanang R Ariefta[1], Eloiza May S Galon[3], Shimaa AES El-Sayed[1,4], Thom Do[1], Lijun Jia[2], Miako Sakaguchi[5], Masahito Asada[1], Yoshifumi Nishikawa[1], Xin Qin[6], Mingming Liu[6]*, Xuenan Xuan[1,7]*

[1]National Research Center for Protozoan Diseases, Obihiro University of Agriculture Veterinary Medicine, Obihiro, Japan; [2]Department of Veterinary Medicine, Agriculture College of Yanbian University, Yanji, China; [3]College of Veterinary Medicine and Biomedical Sciences, Cavite State University, Indang, Philippines; [4]Department of Biochemistry and Molecular Biology, Faculty of Veterinary Medicine, Mansoura University, Mansoura, Egypt; [5]Central Laboratory, Institute of Tropical Medicine (NEKKEN), Nagasaki University, Nagasaki, Japan; [6]School of Basic Medicine, Hubei University of Arts and Science, Xiangyang, China; [7]Research Center for Asian Infectious Diseases, The University of Tokyo, Tokyo, Japan

*For correspondence:
lmm_2010@hotmail.com (ML);
gen@obihiro.ac.jp (XX)

[†]These authors contributed equally to this work

Competing interest: The authors declare that no competing interests exist.

## eLife Assessment

This study presents **valuable** findings with practical and theoretical implications for drug discovery, particularly in the context of repurposing cipargamin CIP for the treatment of Babesia spp. The evidence is **solid** with the methods, data, and analyses broadly supporting the claims. The paper will be of great interest to scientists in drug discovery, computational biology, and microbiology.

**Abstract** Babesiosis is a disease brought on by intraerythrocytic parasites of the genus *Babesia*. Current chemotherapies are accompanied by side effects and parasite relapse. Therefore, it is crucial to develop highly effective drugs against *Babesia*. Cipargamin (CIP) has shown inhibition against apicomplexan parasites, mainly *Plasmodium* and *Toxoplasma*. This study evaluated the growth-inhibiting properties of CIP against *Babesia* spp. and investigated the mechanism of CIP on *B. gibsoni*. The half inhibitory concentration ($IC_{50}$) values of CIP against the in vitro growth of *B. bovis* and *B. gibsoni* were 20.2 ± 1.4 and 69.4 ± 2.2 nM, respectively. CIP significantly inhibited the growth of *B. microti* and *B. rodhaini* in vivo. Resistance was conferred by L921V and L921I mutations in BgATP4, which reduced the sensitivity to CIP by 6.1- and 12.8-fold. The inhibitory potency of CIP against BgATP4-associated ATPase activity was moderately reduced in mutant strains, with a 1.3- and 2.4-fold decrease in BgATP4[L921V] and BgATP4[L921I], respectively, compared to that of BgATP4[WT]. An in silico investigation revealed reductions in affinity for CIP binding to BgATP4[L921V] and BgATP4[L921I] compared to BgATP4[WT]. Resistant strains showed no significant cross-resistance to atovaquone or tafenoquine succinate (TQ), with less than a onefold change in $IC_{50}$ values. Combining CIP with TQ effectively eliminated *B. microti* infection in SCID mice with no relapse, and parasite DNA was not detected by qPCR within 90 days post-infection. Our findings reveal the efficacy of CIP as an antibabesial agent, its limitations as a monotherapy due to resistance development, and the potential of combination therapy with TQ to overcome said resistance and achieve complete parasite clearance.

## Introduction

*Babesia* is an apicomplexan tick-transmitted hemoparasite, which not only impacts the livestock economy but also causes an emerging disease in humans (*Jalovecka et al., 2019*). *Babesia* is a global pathogen that is more prevalent in certain regions such as Asia, Europe, and North America. However, as climate change brings with it higher temperatures and humidity, ticks and their reservoir hosts are anticipated to expand northward for survival and activity (*Gray and Ogden, 2021*). The disease is known as babesiosis, commonly characterized by fever and hemolytic anemia, but chronic infections can be asymptomatic (*Almazán et al., 2022*). The fatality rate ranges from 1% among all cases to 3% among hospitalized cases, and as high as 20% in immunocompromised patients (*Krause, 2019*).

Currently, the most common drugs for the treatment of human babesiosis include a combination of atovaquone (ATO) plus azithromycin (AZI) or clindamycin (CLN) plus quinine (QUI) (*Krause et al., 2021*). Still, these recommended treatments often result in various problems. For instance, multiple mutations in the *B. microti* cytochrome b, which is targeted by ATO, were identified in patients with relapsing babesiosis (*Holbrook et al., 2023*; *Krause et al., 2024*; *Lemieux et al., 2016*; *Marcos et al., 2022*; *Rogers et al., 2023*; *Rosenblatt et al., 2021*; *Simon et al., 2017*). The combination of CLN and QUI is the last resort for patients with severe symptoms (*Kletsova et al., 2017*). Despite its efficacy, this combination can elicit adverse drug reactions (*Vannier and Krause, 2012*). The 8-aminoquinoline analog tafenoquine (TQ) was found to be effective in curing immunocompromised patients experiencing relapsing babesiosis caused by *B. microti* (*Rogers et al., 2023*). However, the efficacy of TQ treatment may vary between individuals and cases, and its contraindication in patients with glucose-6-phosphate dehydrogenase (G6PD) deficiency limits its clinical use (*Chu and Freedman, 2019*). Due to the aforementioned issues, it is undoubtedly urgent to search for compounds that treat infections caused by *Babesia* spp. while simultaneously minimizing the detrimental side effects of antibabesial drugs.

Spiroindolone cipargamin (CIP), a promising antimalarial, has been found to effectively suppress the growth of all strains of *Plasmodium falciparum* and *P. vivax* with potency in the low nanomolar ranges (*Rosling et al., 2018*; *Rottmann et al., 2010*). The assessment of the drug indicated that CIP taken orally had good absorption, a long half-life, and exceptional bioavailability (*Schmitt et al., 2022*). Following oral CIP treatment (30 mg daily for 3 days) in adults with simple *P. falciparum* or *P. vivax* malaria, parasitemia was rapidly cleared in a phase II trial (*Schmitt et al., 2022*). Currently, a clinical trial is being conducted for the intravenous administration of CIP as a treatment for severe malaria patients (*ClinicalTrials, 2024*). CIP was also evaluated for treating toxoplasmosis as a second-line medicine in cases where there are intolerable toxicity issues or allergies to the currently used treatments. Mice infected with *Toxoplasma gondii* that were treated with CIP on the day of infection and the following day had 90% fewer parasites 5 days post-infection (DPI) (*Zhou et al., 2014*).

*Pf*ATP4, a P-type ATPase in *P. falciparum*, functions as a $Na^+/H^+$ transporter critical for maintaining ionic balance (*Spillman et al., 2013*). Mutations in *Pf*ATP4 confer resistance to CIP, which disrupts $Na^+$ homeostasis, leading to increased cytosolic $Na^+$ concentration and various physiological changes, such as cytosolic alkalinization and osmotic swelling (*Mohring et al., 2022*). These mutations reduce the $Na^+$-dysregulating effects of CIP and the resting $Na^+$ levels. While *Pf*ATP4 is essential for *P. falciparum* survival, its homolog in *T. gondii* (*Tg*ATP4) is less critical, as *T. gondii* experiences brief, high $Na^+$ exposure and can survive without *Tg*ATP4 expression (*Lehane et al., 2019*).

Due to its excellent efficacy against other apicomplexan parasites, including *Plasmodium* spp., and its ability to target parasite ATP4—a conserved protein essential for ion homeostasis in apicomplexan organisms—we hypothesize that CIP could also effectively inhibit *Babesia* spp., a related apicomplexan parasite. Therefore, the objectives of this study were to determine whether CIP could inhibit the growth of *Babesia* spp., namely *B. bovis* and *B. gibsoni* in vitro and *B. microti* and *B. rodhaini* in vivo, and identify the inhibitory mechanisms of CIP on *Babesia* parasites.

## Results

### Inhibitory efficacy of CIP on *B. bovis* and *B. gibsoni* in vitro

In vitro efficacy of CIP against *B. bovis* and *B. gibsoni* showed a steep growth inhibition curve with half inhibitory concentration ($IC_{50}$) values of 20.2 ± 1.4 nM (*Figure 1A*) and 69.4 ± 2.2 nM (*Figure 1B*), respectively. The 50% cytotoxic concentration ($CC_{50}$) value of CIP on Madin–Darby canine kidney

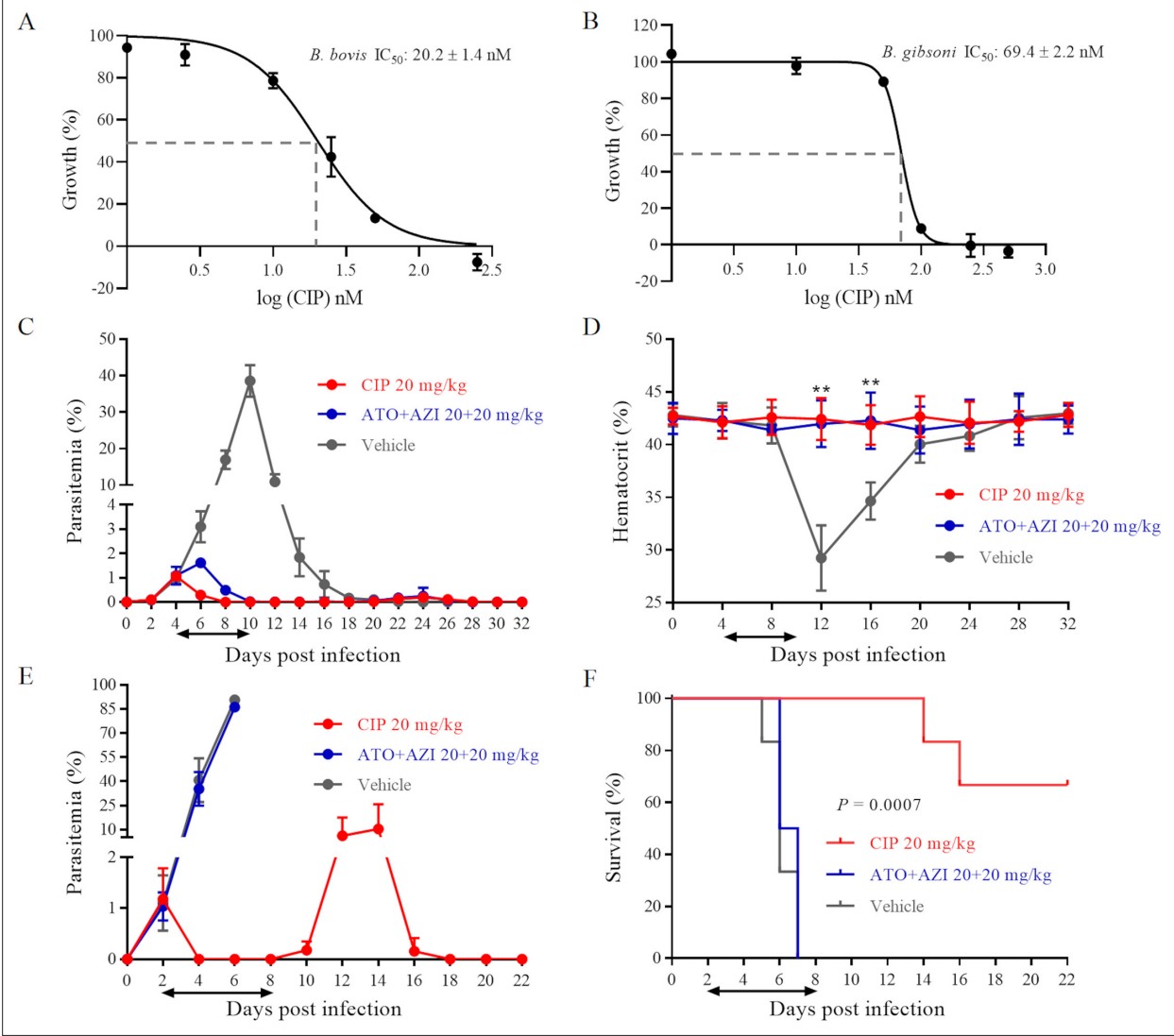

**Figure 1.** Cipargamin (CIP) demonstrates potent inhibition on *Babesia* spp. (**A**, **B**) Dose-dependent growth curve of CIP on *B. bovis* and *B. gibsoni* in vitro. IC$_{50}$: the half maximal inhibitory concentration. (**C**) Inhibitory effects of CIP and atovaquone (ATO) plus azithromycin (AZI) on the proliferation of *B. microti* in BALB/c mice. (**D**) Hematocrit (HCT) values in mice treated with CIP or ATO plus AZI compared with vehicle-treated mice. (**E**) Inhibitory effects of CIP and ATO plus AZI on the proliferation of *B. rodhaini* in BALB/c mice. (**F**) Survival rates of CIP-treated, ATO plus AZI-treated, and vehicle-treated mice. The treatment time is shown by two-way arrows, and significant differences (p < 0.01) between the drug-treated groups (n = 6) and the vehicle-treated control group (n = 6) are indicated by asterisks. The data from one of three individual experiments are expressed as means ± SD. **p < 0.01.

The online version of this article includes the following figure supplement(s) for figure 1:

**Figure supplement 1.** Cytotoxicity assay of CIP on (**A**) Madin-Darby canine kidney (MDCK) cells and (**B**) human foreskin fibroblast (HFF).

(MDCK) cells and human foreskin fibroblasts (HFFs) was 38.7 ± 2.0 and 70.8 ± 4.9 µM (*Figure 1— figure supplement 1*), respectively. Based on these values, the predicted selectivity indices, which reflect the drug's safety and specificity, were calculated to be greater than 500. Furthermore, at a concentration of 100 µM, CIP exhibited a low erythrocyte hemolysis rate of 0.11 ± 0.03% (data not shown).

## CIP effect on *B. microti* and *B. rodhaini* infections in vivo

Concurrently, CIP showed effective inhibition on *B. microti* and *B. rodhaini* in vivo. The parasitemia of *B. microti*-infected BALB/c mice increased dramatically in the vehicle-treated control group and peaked at 10 DPI (38.55 ± 4.32%) (*Figure 1C*). On the other hand, 7 days of treatment with CIP (20 mg/kg) or ATO plus AZI administered orally resulted in a significantly lower peak parasitemia,

1.06 ± 0.20% and 1.61 ± 0.20%, respectively (*Figure 1C*). Hematocrit (HCT) variations were tracked every 4 days as an indicator of anemia in *B. microti*-infected mice. The vehicle-treated group showed a drop in HCT levels at 12 and 16 DPI (p < 0.01) (*Figure 1D*). No significant reduction in HCT levels was observed in the CIP-treated group or the ATO plus AZI-treated group (*Figure 1D*). This indicates that the administration of CIP could control *B. microti* infection and prevent anemia from developing in *B. microti*-infected mice.

BALB/c mice infected with *B. rodhaini* treated with sesame oil or ATO plus AZI showed high parasitemia, 90.73 ± 1.97%, and 86.23 ± 3.06%, respectively (*Figure 1E*), and all mice died within 7 DPI (*Figure 1F*). CIP treatment in *B. rodhaini*-infected mice precluded the emergence of parasitemia for the following 8 days (*Figure 1E*), which led to 66.67% of mice surviving the challenge infection (*Figure 1F*). At 12 DPI, parasites had recurred in all CIP-treated *B. rodhaini*-infected mice (10.32 ± 15.51%), which were eventually cleared as indicated by undetectable parasites at 18 DPI (*Figure 1E*).

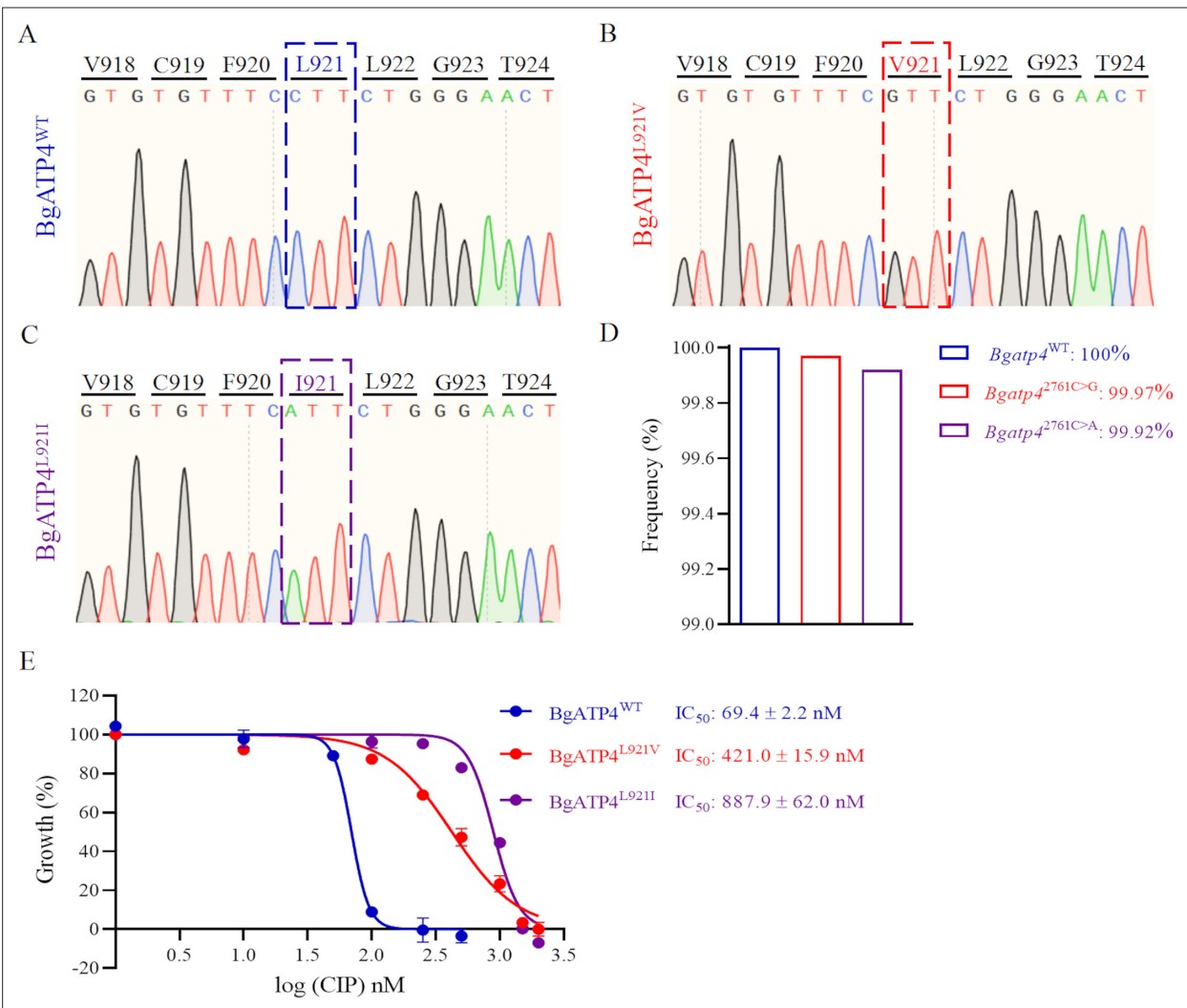

**Figure 2.** Mutations in BgATP4 mediate cipargamin (CIP) resistance. (**A–C**) Representative sequencing chromatogram of wild-type and resistant parasites from CIP-treated *B. gibsoni*. The resistant parasite genomic DNA is extracted from blood samples after a 60-day treatment. The BgATP4 gene was amplified and sequenced using the DNA. (**D**) Genes of high-frequency sequence variants detected by next-generation sequencing (NGS). (**E**) Dose-dependent growth curve of BgATP4$^{WT}$, BgATP4$^{L921V}$, and BgATP4$^{L921I}$ in vitro. The data from one of three individual experiments are expressed as means ± SD.

The online version of this article includes the following figure supplement(s) for figure 2:

**Figure supplement 1.** Multiple sequence alignment of ATP4 in different species.

## Identification of *B. gibsoni* ATP4 mutation in CIP-resistant strains

After being exposed to CIP at increasing concentrations up to 10 times the $IC_{50}$, the resistant parasites in two of the culture wells were able to regrow. We sequenced the *B. gibsoni* ATP4 gene from the wild-type and two resistant strains. The wild-type strain has a C at nucleotide 2761, which translates to leucine (*Figure 2A*). In one resistant strain, a single-nucleotide variant in BgATP4 with a substitution at position 2761 (from C to G) was found—a nonsynonymous coding change from leucine to valine (L921V) (*Figure 2B*). In another resistant strain, the mutation occurred in the same position. However, the nucleotide substitution was from C to A, and the coding changed from leucine to isoleucine (L921I) (*Figure 2C*). Next-generation sequencing (NGS) revealed that for $Bgatp4^{2761C>G}$, 99.97% of 7,960 reads were G at nucleotide 2761, and for $Bgatp4^{2761C>A}$, 99.92% of 7862 reads were A at nucleotide 2761 (*Figure 2D*). $BgATP4^{L921V}$ and $BgATP4^{L921I}$ lines were tested for their susceptibility to CIP and had $IC_{50}$ values of 421.0 ± 15.9 and 887.9 ± 62.0 nM, respectively (*Figure 2E*). These findings demonstrate a 6.1- and 12.8-fold reduction in CIP sensitivity of the resistant parasite lines $BgATP4^{L921V}$ and $BgATP4^{L921I}$.

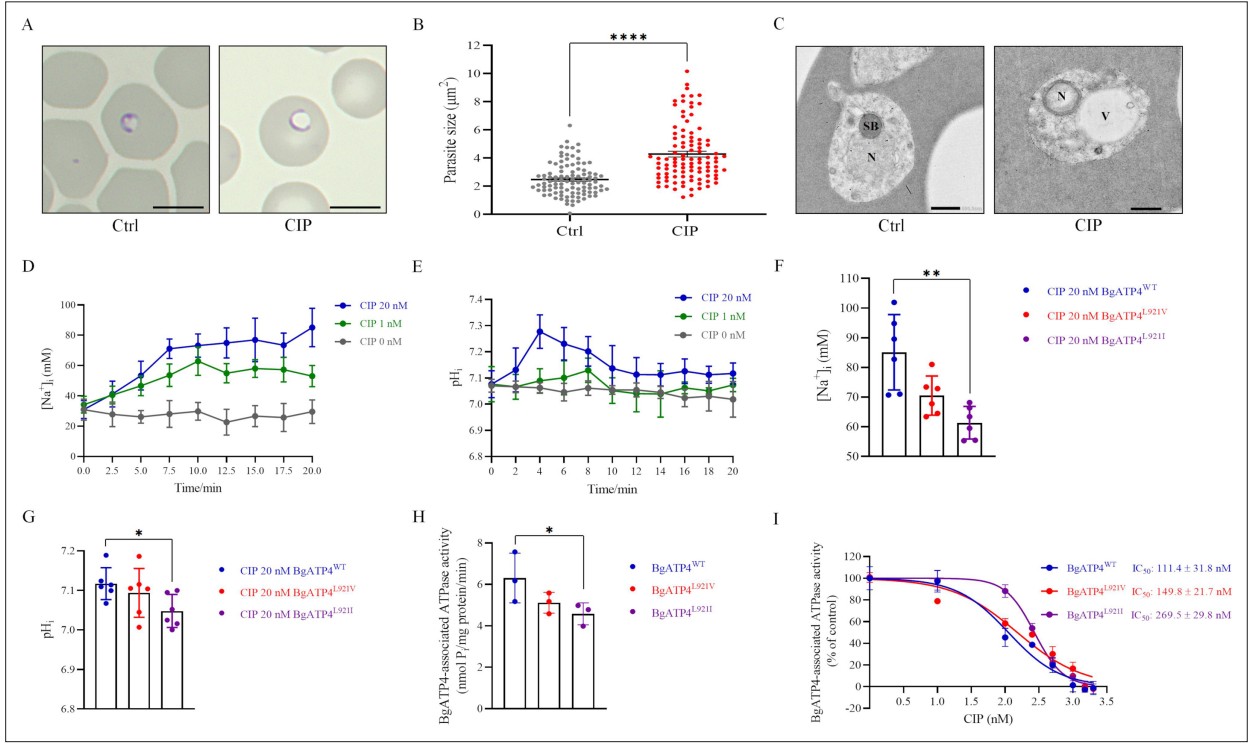

**Figure 3.** Mechanistic basis for resistance to cipargamin (CIP) conferred by the L921V and L921I mutations in BgATP4. (**A**) Untreated and CIP-treated parasite morphology after incubation for 72 hr. Scale bar: 5 μm. (**B**) Sizes of 100 parasites in two groups measured with ImageJ software in panel A. Statistically significant differences between the means of variables determined by *t*-test. ****$p < 0.0001$. (**C**) Transmission electron microscopy (TEM) of untreated and CIP-treated parasite. N, nucleus; SB, spherical body; V, vacuole. Scale bar: 500 nm. (**D**) $[Na^+]_i$ concentrations after the addition of CIP in the $BgATP4^{WT}$ line. Representative traces from the experiment that highlight the impact of adding 20 nM CIP (blue), 1 nM CIP (green), or 0 nM CIP (grey) on the concentration $[Na^+]_i$ of the $BgATP4^{WT}$ line. (**E**) Alkalinization of $pH_i$ in $BgATP4^{WT}$ line upon addition of the ATP4 inhibitor. (**F**) Addition of 20 nM CIP to the wild-type and resistant parasite lines results in different $[Na^+]_i$ concentrations. (**G**) Addition of 20 nM CIP to the wild-type and resistant parasite lines results in different $pH_i$ concentrations. (**H**) Data acquired in the low $Na^+$ condition (containing only the 2 mM $Na^+$ introduced upon the addition of 1 mM $Na_2ATP$) was subtracted from data obtained in the high $Na^+$ condition to determine the ATPase activity related to the BgATP4 proteins. (**I**) Dose-dependent BgATP4-associated ATPase activity curve of $BgATP4^{WT}$, $BgATP4^{L921V}$, and $BgATP4^{L921I}$ in vitro. ATPase activity was determined at pH 7.2 in the presence of 150 mM $Na^+$ and 1 mM $Na_2ATP$. Each value represents the mean ± SD derived from a minimum of three biological replicates. *$p < 0.05$; **$p < 0.01$.

The online version of this article includes the following figure supplement(s) for figure 3:

**Figure supplement 1.** Proposed mechanism of inhibition of cipargamin (CIP) on wild-type and mutant parasite-infected erythrocytes.

## The effect of CIP on BgATP4$^{WT}$, BgATP4$^{L921V}$, and BgATP4$^{L921I}$ function

Microscopic observation of thin blood smears was performed to determine the morphological changes of *B. gibsoni* exposed to CIP. The CIP-treated parasites became swollen after incubation with the drug for 72 hr (*Figure 3A*). For both the treatment and control groups, one hundred parasites were measured. The mean size of treated parasites was notably bigger than the parasites in the untreated group (p < 0.0001) (*Figure 3B*). Significant vacuolization was observed in the cytoplasm of parasites in the CIP-treated group, as revealed by transmission electron microscopy (TEM). Despite this, the nuclear membrane structure and parasitic membranes remained intact until the parasites were completely destroyed (*Figure 3C*). The addition of the ATP4 inhibitor CIP resulted in a time-dependent increase in the concentrations of $[Na^+]_i$ in wild-type *B. gibsoni*, with improved signal-to-noise ratios at the higher drug concentration of 20 nM (*Figure 3D*). We observed that the $Na^+$ concentrations in both BgATP4$^{L921V}$ and BgATP4$^{L921I}$ lines were lower when compared with those of the control BgATP4$^{WT}$ line after being exposed to 20 nM CIP for 20 min, with a significantly lower $Na^+$ concentration in BgATP4$^{L921I}$ (p = 0.0087) (*Figure 3F*). We also demonstrated here that the addition of CIP in wild-type *B. gibsoni* caused an increase in the cytosolic pH (*Figure 3E*). Specifically, 4 min after the drug was added, the average pH of the 20 nM CIP group reached as high as 7.278, while the 1 nM CIP group reached 7.089 and the untreated group reached 7.062 (*Figure 3E*). The pH values of the 20 nM CIP group were consistently higher than those of the other two groups, although declining with time (*Figure 3E*). In resistant lines, a 20-min exposure to 20 nM CIP caused small changes in the pH values compared to the wild-type line, with the BgATP4$^{L921I}$ line (7.048 ± 0.042) having a notably lower pH value (p = 0.0229) (*Figure 3G*).

## Sensitivity of BgATP4-associated ATPase activity to CIP in BgATP4$^{WT}$, BgATP4$^{L921V}$, and BgATP4$^{L921I}$

The BgATP4-associated ATPase activity in erythrocytes infected with BgATP4$^{WT}$ (6.31 ± 1.20 nmol Pi/mg protein/min), measured in the presence of 150 mM $Na^+$, was higher than those observed in BgATP4$^{L921V}$ (5.11 ± 0.50 nmol Pi/mg protein/min) and BgATP4$^{L921I}$ (4.58 ± 0.53 nmol Pi/mg protein/min) (p = 0.04) (*Figure 3H*).

We further investigated the concentration-dependent inhibition of BgATP4-associated ATPase activity by CIP in wild-type and mutant parasites. In membranes prepared from *B. gibsoni*, CIP inhibited BgATP4-associated ATPase activity with $IC_{50}$ values of 111.4 ± 31.8 nM for BgATP4$^{WT}$, 149.8 ± 21.7 nM for BgATP4$^{L921V}$, and 269.5 ± 29.8 nM for BgATP4$^{L921I}$. The potency of CIP in inhibiting BgATP4-associated ATPase activity was reduced by 1.3- and 2.4-fold in membranes prepared from BgATP4$^{L921V}$ and BgATP4$^{L921I}$, respectively, compared to BgATP4$^{WT}$ (*Figure 3I*).

## Multiple sequence alignment of *Babesia* ATP4 and molecular docking

The whole amino acid sequence of *B. gibsoni* ATP4 (GenBank: KAK1443404.1) shared identity values of 29.75%, 49.40%, 49.67%, 62.21%, and 52.47% with *Homo sapiens* ATP4 (GenBank: NM_000704.3), *P. falciparum* ATP4 (GenBank: PF3D7_1211900), *T. gondii* ATP4 (GenBank: XP_018635122.1), *B. bovis* ATP4 (PiroplasmaDB: BBOV_IV010020), and *B. microti* ATP4 (GenBank: BMR1_03g01005), respectively (*Figure 2—figure supplement 1*).

The pLDDT (predicted Local Distance Difference Test) value of BgATP4$^{WT}$ prediction was 80.7 using Colab-fold. Multiple potential binding sites for CIP were revealed by blind docking throughout the whole protein surface (*Figure 4—figure supplement 1*). CIP binds in close proximity to L921, as demonstrated by focused docking on this area (*Figure 4A*). The contribution of each residue to the predicted binding affinity in either mutant structure was reduced; the precise values of BgATP4$^{WT}$ (*Figure 4B*), BgATP4$^{L921V}$ (*Figure 4C*), and BgATP4$^{L921I}$ (*Figure 4D*) were −6.43, −6.40, and −6.26 kcal/mol, respectively. The interactions of CIP from each docking simulation are shown in *Supplementary file 3*.

## Cross-resistance of BgATP4$^{L921V}$ and BgATP4$^{L921I}$ mutants to ATO and TQ

The BgATP4$^{WT}$, BgATP4$^{L921V}$, and BgATP4$^{L921I}$ lines were tested for their susceptibility to ATO and TQ. The $IC_{50}$ values for these lines were 380.1 ± 3.1, 412.9 ± 5.4, and 360.3 ± 7.9 nM, respectively, for ATO (*Figure 5A*), and 39.5 ± 0.4, 29.9 ± 1.4, and 39.3 ± 0.3 μM, respectively, for TQ (*Figure 5B*). The $IC_{50}$ values for both ATO and TQ in the resistant strains showed only slight changes compared to the

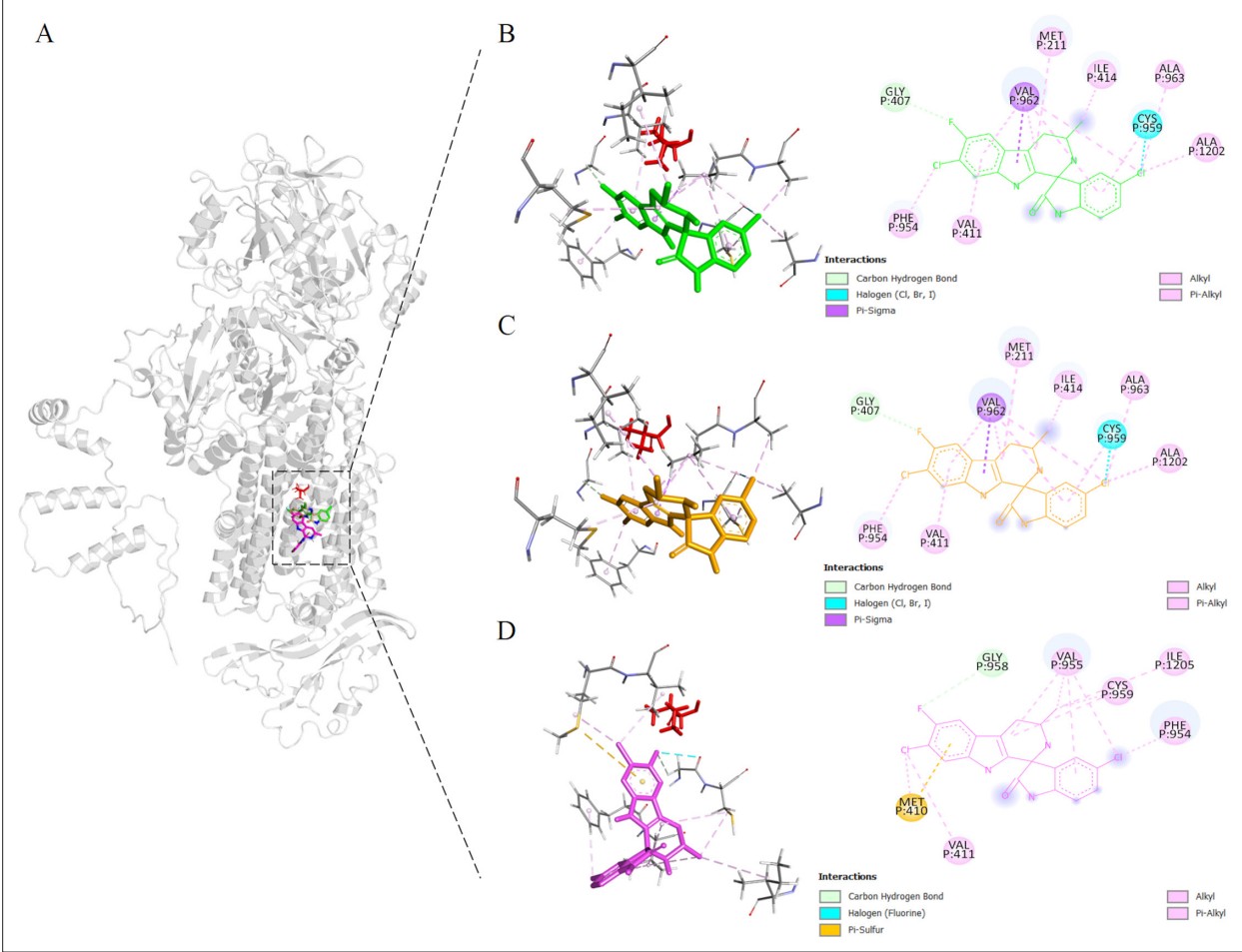

**Figure 4.** Binding sites proximal to BgATP4 residue 921 predicted by molecular docking. (**A**) The lowest energy poses for cipargamin (CIP) were located in reference to the whole protein structure, docking against the WT (green), L921V (yellow), and L921I (pink) mutant BgATP4. The side chain of L921 is also shown in a red stick at its position. (**B–D**) The zoomed views of the binding locations of CIP.

The online version of this article includes the following figure supplement(s) for figure 4:

**Figure supplement 1.** Binding sites for cipargamin (CIP) found by Gnina search across the entire surface of the protein.

wild-type strain, with less than a onefold difference. This minimal variation suggests that the resistant strain has a mild alteration in susceptibility to ATO and TQ, but not enough to indicate strong resistance or significant cross-resistance.

## Combination treatment of CIP plus TQ in SCID mice with *B. microti* infection

The parasitemia of *B. microti*-infected SCID mice in the vehicle group increased dramatically, peaking at 10 DPI (77.03 ± 2.45%) (*Figure 5C*). Although the trend showed a subsequent decline, the parasitemia remained around 50% until the mice were euthanized at 90 DPI. Treatment with CIP (20 mg/kg) initially reduced parasitemia to undetectable by 18 DPI. However, a relapse occurred, with parasitemia increasing rapidly and stabilizing at levels comparable to the vehicle group during the subsequent observation period. Notably, no mutations were detected in the relapsed *B. microti* parasites from SCID mice (data not shown). In contrast, parasitemia was completely cleared in the TQ and CIP plus TQ groups by 8 DPI and 6 DPI, respectively, with no parasites observed under the microscope thereafter. qPCR analysis at 90 DPI detected parasite DNA in all groups except the CIP plus TQ group (*Figure 5D*).

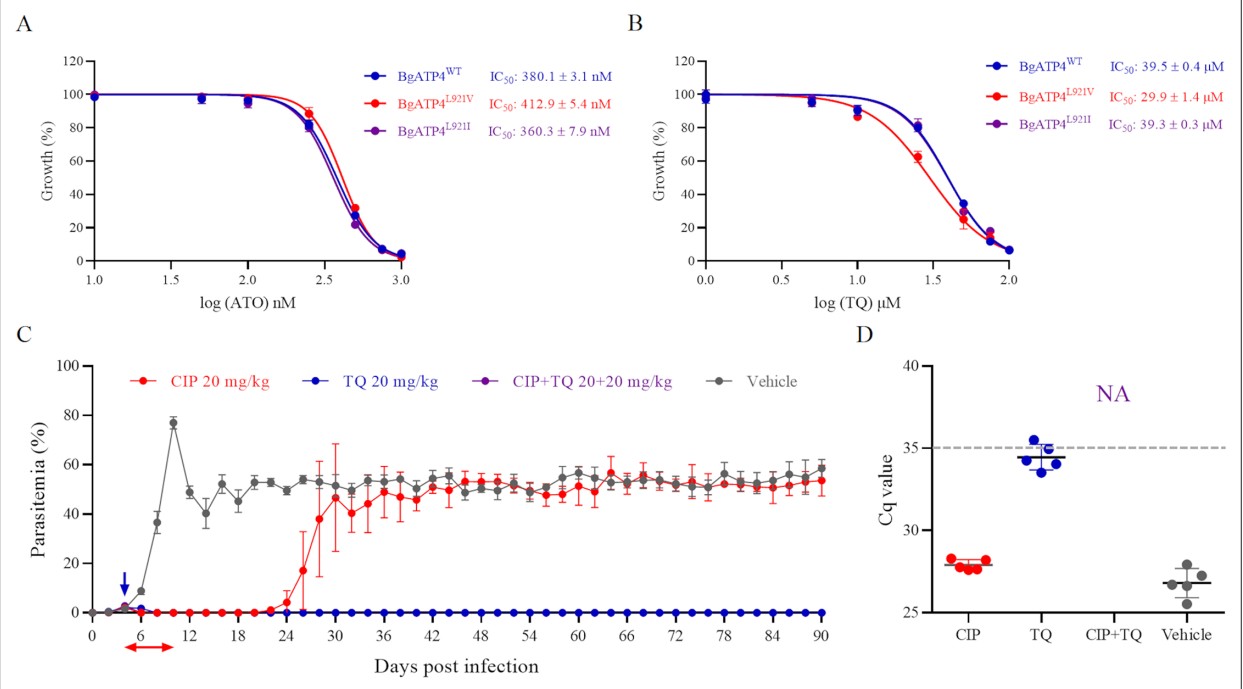

**Figure 5.** Cross-resistance between ATO and TQ in resistant parasites and combination therapy based on cipargamin (CIP) plus TQ. (**A**) Dose-dependent growth curve of BgATP4[WT], BgATP4[L921V], and BgATP4[L921I] by ATO treatment in vitro. (**B**) Dose-dependent growth curve of BgATP4[WT], BgATP4[L921V], and BgATP4[L921I] by TQ treatment in vitro. (**C**) Inhibitory effects of CIP plus TQ on the proliferation of *B. microti* in SCID mice (n = 5 per group). Treatment started at 4 DPI: CIP was given at 20 mg/kg once daily for 7 days, TQ was administered as a single 20 mg/kg dose, and the combination group received both treatments (CIP at 20 mg/kg once daily for 7 days plus a single dose of TQ at 20 mg/kg). (**D**) Parasite DNA was detected by qPCR on genomic DNA extracted from blood collected from untreated and treated SCID mice infected with *B. microti* at 90 DPI. A dotted gray line across the graph represents the average cut-off Cq value. Cut-off Cq ≤35 was considered positive, while Cq >35 or no amplification was considered negative. Each value represents the mean ± SD derived from a minimum of five biological replicates. NA indicates no amplification.

## Discussion

The repositioning of antimalarial drugs is critical in developing novel strategies for treating babesiosis. CIP is a novel compound that inhibits *Plasmodium* development by targeting ATP4 and has been extensively tested in Phase 1 and 2 clinical trials (*Qiu et al., 2022*). Since ATP4 is conserved across apicomplexans, including *Babesia* species, our study aimed to repurpose the antimalarial CIP by assessing its efficacy on *Babesia* species (*Lehane et al., 2019*). The $IC_{50}$ values of CIP against *B. bovis* and *B. gibsoni* in vitro were lower than the previously reported $IC_{50}$ value of TQ (*Carvalho et al., 2020*; *Ji et al., 2022b*) and ATO against *B. gibsoni* (*Matsuu et al., 2004*). The present investigation also demonstrated that the inhibitory effects of CIP on *B. microti*-infected BALB/c mice were comparable to that of ATO plus AZI, the combination recommended by the CDC in the United States. CIP was also proven to protect mice from the deadly *B. rodhaini* infection, with a survival rate of up to 67%, as recorded in the current trial. These results suggest that CIP may be a potential chemotherapy candidate for babesiosis.

In a previous study, the *P. falciparum* Dd2 strain that acquired resistance to CIP carried the G358S mutation in the *Pf*ATP4 protein, a *P. falciparum* P-type Na⁺ ATPase (*Qiu et al., 2022*). The vital protein ATP4 is found in the parasite plasma membrane and is specific to the subclass of apicomplexan parasites (*Mohring et al., 2022*). To date, in vitro evolution experiments using CIP have produced at least 18 parasite lines with various *Pf*ATP4 mutations (*Lee and Fidock, 2016*; *Rottmann et al., 2010*). In another study involving *T. gondii*, a cell line that carried the mutation G419S in the *Tg*ATP4 gene was 34 times less susceptible to CIP than that of *Tg*ATP4[WT] (*Qiu et al., 2022*). It follows that there is a significant possibility that resistant *Babesia* parasites will emerge from CIP exposure. In this study, we successfully produced two CIP-resistant strains using six independent selections. The newly discovered L921V and L921I mutations in BgATP4 decreased the CIP sensitivity by 6.1 and 12.8 times, respectively. We provided compelling evidence that natural variety mutations L921V and L921I in

BgATP4 significantly affected the protein's susceptibility to CIP inhibition, albeit the mutational sites were different from those of *P. falciparum* and *T. gondii*. Based on our observation, the growth and generation rates of the mutant strains are comparable to those of the wild-type strain.

Although ATP4 was initially recognized as a $Ca^{2+}$ transporter (**Krishna et al., 2001**), current evidence suggests that ATP4 functions as an ATPase for exporting $Na^+$ while importing $H^+$ (**Mohring et al., 2022**), as well as causing a variety of other physiological perturbations including an increase in the volume of parasites and infected erythrocytes due to the osmotic impact of the $[Na^+]_{cyt}$ increase (**Dennis et al., 2018**), a decline in cholesterol. Extrusion from the parasite plasma membrane as a result of the increase in $[Na^+]_{cyt}$ (**Das et al., 2016**), and an intensified rigidity of erythrocytes infected with ring-stage parasites (**Zhang et al., 2016**). In this study, the CIP-exposed wild-type *B. gibsoni* became swollen, which was identical to a prior study on *P. falciparum* (**Dennis et al., 2018**). Interestingly, TEM analysis of parasites incubated with CIP revealed ultrastructural alterations characterized by significant vacuole formation in the cytoplasm, a hallmark of stress or early stages of cell death. These changes were similar to those observed in *B. bovis* treated with clotrimazole and ketoconazole, which ultimately led to growth inhibition or parasite death (**Bork et al., 2003**). Despite the vacuolization, the structural integrity of the nucleus and membranes was maintained, suggesting a gradual process where vacuolization precedes complete destruction of the parasite. One explanation is the swelling of the isolated parasites, which can be ascribed to the osmotic consequences of $Na^+$ uptake and is contingent upon the presence of $Na^+$ in the external environment.

The output of $Na^+$ and input of $H^+$ diminished upon CIP-induced inhibition of *Pf*ATP4, and the continued outflow of $H^+$ via V-type $H^+$-ATPase led to an alkalinization that ultimately killed the parasites (**Spillman et al., 2013**). Our capacity to measure a time-dependent increase in the concentration of $[Na^+]_i$ and pH value for wild-type *B. gibsoni* facilitated us to gain insight into the basic mechanisms of CIP on BgATP4$^{WT}$ function. Furthermore, our results corroborate internal alkalinization as the main factor in *Babesia* death and support our hypothesis that the swollen isolated parasites were produced by $Na^+$ absorption. To explore further how mutations in BgATP4 are associated with the upregulation of the parasite's $[Na^+]_i$ and $[H^+]_i$, we tested two BgATP4-mutant lines that were chosen previously with BgATP4 inhibitors (BgATP4$^{L921V}$ and BgATP4$^{L921I}$). Due to the presence of L921V and L921I mutations in drug-resistant strains of BgATP4, the concentration of $Na^+$ did not increase as much as it would have in the wild-type strain following the addition of 20 nM CIP for 20 min. As intraerythrocytic alkalinization rises, the same outcome happens. From these results, we deduced how natural mutations in BgATP4 may affect ATP4 inhibitor susceptibility by dysregulating $H^+$ and $Na^+$ balance, which helped parasites survive in a relatively high concentration of CIP. An illustration of the putative processes for regulating $Na^+$ and $H^+$ in erythrocytes infected with the *Babesia* parasite is presented in *Figure 3—figure supplement 1*.

In a previous study's isolated membrane assay, CIP was found to be more effective at inhibiting parasite development than *Pf*ATP4-associated ATPase activity, and reduced sensitivity of *Pf*ATP4-associated ATPase activity to the medication was associated with a decrease in parasite susceptibility to CIP (**Rosling et al., 2018**). In our study, the higher resistance to CIP-mediated inhibition of parasite proliferation observed in BgATP4$^{L921I}$ compared to BgATP4$^{L921V}$ correlated with a higher resistance of BgATP4-associated ATPase activity to the drug. These findings provide additional evidence that the ATPase activity measured in this study corresponds to the activity of the BgATP4 protein.

According to studies using the *Pf*ATP4 model for molecular docking, the G358S mutation results in a steric clash that lowers CIP's binding affinity (**Qiu et al., 2022**). The results from the current study were similar to the *Pf*ATP4 model. The molecular docking was constructed using a ColabFold model of wild-type BgATP4, which predicted the binding mode and affinity between ATP4 protein and the ligand to provide a possible mechanistic explanation. It suggested that the L921V mutation caused changes at the atomic level, whereas the L921I mutation created a steric clash that reduced the binding affinity of CIP. The predicted affinity score of the L921I mutation was lower than that of the L921V mutation. Thus, it is possible that CIP had a weaker binding to BgATP4$^{L921I}$ than to BgATP4$^{L921V}$. These findings are consistent with the results obtained from measuring the IC$_{50}$ of the drug against parasites with the L921V and L921I mutations.

While CIP shows promise as an antibabesial agent, the emergence of resistance emphasizes the importance of rational drug use and the development of combination therapies. Our results demonstrate that CIP did not exhibit cross-resistance with ATO or TQ, as IC$_{50}$ values for ATO and TQ in

resistant strains showed minimal changes (<1-fold) compared to the wild-type strain. This observation suggests that CIP could be effectively combined with other antibabesial drugs to reduce the risk of resistance development. Previous studies have also reported that a single dose of TQ may not be sufficient to prevent parasite relapse in immunocompromised hosts, recommending either repeated TQ administration or combination therapy with other antibabesial agents to address this issue (**Mordue and Wormser, 2019**). Supporting this approach, our study demonstrates that the combination of CIP plus TQ effectively cleared *B. microti* infections in SCID mice, with no detectable parasitemia or DNA observed at 90 DPI, indicating complete parasite elimination.

This preclinical finding underscores the potential of the CIP plus TQ combination as a viable treatment strategy for babesiosis, especially in cases where monotherapy is ineffective. Further studies are needed to fully explore the pharmacokinetics, long-term efficacy, and potential side effects of such combinations in clinical settings, ultimately facilitating optimized therapeutic strategies.

In summary, our findings provide a comprehensive understanding of CIP's efficacy, mechanism of resistance, and identifying strategies to enhance its therapeutic potential. The combination of CIP with drugs like TQ offers a promising approach to combat babesiosis and mitigate the development of resistance.

# Materials and methods

## Parasite culture

The parasites *B. gibsoni* Oita strain and *B. bovis* Texas strain were in vitro cultured in 24-well plates and maintained in an atmosphere of 5% $CO_2$ and 5% $O_2$ at 37°C (**Liu et al., 2018**). For the in vivo studies, *B. microti* Peabody mjr strain-(ATCC PRA-99) and *B. rodhaini* Australia strain-infected RBCs (iRBCs), which were collected and diluted with phosphate-buffered saline when the parasitemia levels in the donor mice reached ~20% and 50%, respectively, and were intraperitoneally injected into 6-week-old female BALB/c mice. Each BALB/c mouse was infected with $1.0 \times 10^7$ *B. microti* or *B. rodhaini* iRBCs for the in vivo trials (**Ji et al., 2022a**).

## In vitro cytotoxicity of CIP and hemolysis rate in canine erythrocytes

HFFs and MDCK cells were obtained from the American Type Culture Collection (ATCC, catalog number CRL-2522 and CCL-34). Both cell lines were authenticated using short tandem repeat profiling conducted by ATCC to confirm their genetic identities. Additionally, both HFF and MDCK cells were tested and confirmed the absence of mycoplasma contamination by fluorescent Hoechst staining. Cell lines were maintained at 37°C under an atmosphere of 5% $CO_2$ and 5% $O_2$, and the cytotoxic effect of CIP (MedChem Express, Tokyo, Japan) was assessed using a cell viability assay by CCK-8 (Dojindo, Japan) as described previously (**Li et al., 2023**). The selectivity index is calculated as the ratio between the $IC_{50}$ and the $CC_{50}$ values.

Canine erythrocytes were collected from healthy beagle dogs raised in NRCPD and stocked in Vega y Martinez (VYM) phosphate-buffered saline solution at 4°C (**Vega et al., 1985**). A canine erythrocyte hemolysis assay was performed at concentrations of 0.1, 1, 5, 10, 25, 50, and 100 µM as previously described (**Ariefta et al., 2022**).

## Evaluation of the efficacy of CIP against *Babesia* parasites in vitro

The efficacy of CIP against *B. gibsoni* and *B. bovis* was determined using a fluorescence assay, as previously described (**Guswanto et al., 2014**). The $IC_{50}$ values were determined from the fluorescence values and by non-linear regression analysis (curve fit) in GraphPad Prism 9 (GraphPad Software Inc, USA).

## Chemotherapeutic effects of CIP against *Babesia* infections in vivo

CIP was evaluated on *B. microti*- and *B. rodhaini*-infected mice, as previously described (**Ndayisaba et al., 2021**; **Tuvshintulga et al., 2022**). When *B. microti*- and *B. rodhaini*-infected mice had a 1% average parasitemia at 4 and 2 DPI, respectively, the drug treatments were administered and continued for 7 days. Three groups of *B. microti*-infected mice were administered different treatments. The CIP group (*n* = 6) and the ATO plus AZI group (*n* = 6) were orally treated with 20 mg/kg CIP and 20 mg/kg ATO plus 20 mg/kg AZI (Sigma, Tokyo, Japan), respectively. All drugs were prepared in sesame

oil. The infected mice of the vehicle group ($n = 6$) orally received 0.2 ml of sesame oil as the control. Eighteen mice infected with *B. rodhaini* were likewise placed into three groups for treatment, and they received the same treatments as the mice infected with *B. microti*. A light microscope (Nikon, Japan) and a hematology analyzer (Celltac α MEK-6450, Nihon Kohden Corporation, Tokyo, Japan) were used to assess the parasitemia and HCT levels every 2 and 4 days, respectively.

### Selection of CIP-resistant *B. gibsoni* in vitro

Selections were initiated by exposing six independent flasks, each containing 10 µl ($5 \times 10^6$) *B. gibsoni* iRBCs mixed with 40 µl ($4 \times 10^8$) RBCs into a 450-µl culture medium, which contained increasing concentrations of CIP: 5, 10, 20, and 30–694 nM ($10 \times IC_{50}$). The medium containing CIP was replaced daily until parasites treated with $10 \times IC_{50}$ CIP reached multiplication rates that were approximately comparable to those of the untreated controls (*Hwang et al., 2010*). Then, to evaluate the decreased sensitivity to CIP, $IC_{50}$ values of resistant strains were determined by nonlinear regression using the GraphPad Prism software.

### Detection of *B. gibsoni* ATP4 gene mutations

The genomic DNA of the mutant parasites was extracted and sequenced (*Ji et al., 2022a*). The primer sets used for sequencing are listed in *Supplementary file 1*. The single-nucleotide variants were identified by pairwise alignment to the BgATP4$^{WT}$ sequence (GenBank: KAK1443404.1). NGS was performed to detect the ratio of wild-type to mutant parasites by using the Illumina NovaSeq6000 sequencing platform (*Jeon et al., 2021*).

### Morphological changes in CIP-treated in vitro cultured *B. gibsoni*

A microscopy assay was used to detect the morphological changes of wild-type *B. gibsoni* after exposure to 50 nM CIP for three consecutive days (*Liu et al., 2021*; *Tayebwa et al., 2018*). At 4 days post-treatment, ImageJ software was used to measure the sizes of 100 randomly selected parasites in the CIP-treated group and the control group on Giemsa-stained blood smears. After treating the parasites as described above, iRBCs were fixed in GA fixation buffer (2% glutaraldehyde in 0.1 M sodium cacodylate buffer containing 1 mM $CaCl_2$ and 1 mM $MgCl_2$), then stored in rinse buffer (0.1 M sodium cacodylate buffer containing 1 mM $CaCl_2$ and 1 mM $MgCl_2$) at 4°C for TEM analysis (*Hidayati et al., 2023*).

### Parasites [Na$^+$]$_i$ and pH$_i$ measurements

For both [Na$^+$]$_i$ and pH$_i$ measurements, the wild-type and mutant parasites were initially separated from erythrocytes by treatment with saponin (0.05% [wt/vol]) for 5 min (*Saliba and Kirk, 1999*). The Na$^+$-sensitive fluorescent dye SBFI (Thermo Fisher Scientific; product S1263) was used to quantify intracellular sodium ([Na$^+$]$_i$). Saponin-isolated parasites (at $1 \times 10^8$ parasites/ml) were loaded with SBFI (5 µM; in the presence of 0.02% wt/vol Pluronic F127) for 1 hr at room temperature (RT) (*Spillman et al., 2013*). Thereafter, SBFI-loaded parasites were resuspended in physiological saline (120 mM NaCl, 5 mM KCl, 25 mM HEPES, 20 mM D-glucose, and 1 mM $MgCl_2$ [pH 7.1]) at RT in the presence or absence of CIP. A 96-well microtiter plate was filled with around 200 µl of parasite suspension per well. The dye-loaded cells fluoresced at 515 nm after being stimulated at 340 and 380 nm. Parasites loaded with SBFI were suspended in calibration buffers containing [Na$^+$] values ranging from 0 to 140 mM (pH 7.1) to establish calibration curves (*Diarra et al., 2001*).

The cytosolic pH of wild-type and mutant strains was measured using the pH-sensitive fluorescent dye BCECF [2′,7′-bis-(2-carboxyethyl)-5-(and-6)-carboxyfluorescein] (Biotium; product 51011). The BCECF was added to saponin-isolated parasites by suspension ($1 \times 10^8$ parasites/ml) and incubated for 20 min at 37°C in RPMI-1640 culture medium (Gibco, USA) (*Saliba and Kirk, 1999*). Thereafter, the parasites loaded with dye were rinsed thrice ($12,000 \times g$, 1 min) in RPMI-1640 culture medium, then resuspended in physiological saline (as previously mentioned) with or without various concentrations of CIP. A 96-well microtiter plate was filled with around 200 µl of parasite suspension per well. The dye-loaded cells fluoresced at 520 nm after being stimulated at 440 and 490 nm. A pH calibration was carried out for each experiment (*Mohring et al., 2022*).

### Measurements of membrane ATPase activity

Membranes from isolated *B. gibsoni* were prepared by lysing the parasites in ice-cold deionized water containing 7 × Protease Inhibitor Cocktail Tablets (Roche, Germany). The membrane preparation was

then washed three times with ice-cold deionized water, with protease inhibitors included in the first two washes (*Qiu et al., 2022*). Protein concentrations in the membrane samples were determined using a Bradford assay. The production of inorganic phosphate (Pi) from ATP hydrolysis was measured using the Malachite Green Phosphate Assay (BioAssay Systems, USA). Membrane preparations were diluted in either a high $Na^+$ solution (final reaction conditions: 150 mM NaCl, 20 mM KCl, 2 mM $MgCl_2$, 50 mM Tris, pH 7.2) or a $Na^+$-free solution (final reaction conditions: 150 mM choline chloride, 20 mM KCl, 2 mM $MgCl_2$, 50 mM Tris, pH 7.2) to achieve a final protein concentration of 40 µg/ml. CIP was added at the concentrations specified in the respective figure legends. Reactions were conducted according to the manufacturer's protocol provided with the assay kit.

## Multiple ATP4 sequence alignment and molecular docking

*B. gibsoni* ATP4 (GenBank: KAK1443404.1), *B. bovis* ATP4 (PiroplasmaDB: BBOV_IV010020), *B. microti* ATP4 (GenBank: BMR1_03g01005), *T. gondii* ATP4 (GenBank: XP_018635122.1), and *H. sapiens* ATP4 (GenBank: NM_000704.3) sequences were obtained by a homology search using *P. falciparum* ATP4 (GenBank: PF3D7_1211900). Sequence alignment was analyzed using MUSCLE in Jalview v2.11.3.2 software and BLAST (http://www.ncbi.nlm.nih.gov/BLAST/).

AlphaFold was used to predict the structure of BgATP4$^{WT}$ (*Jumper et al., 2021*). Using the GROMACS 2021 Molecular Dynamics package, the energy minimization was carried out following the model's generation (*Lindahl and Hess, 2021*). The mutations were produced by using CHARMM-GUI PDB reader (*Jo et al., 2014*). PyMOL (version 2.0 Schrödinger, LLC) was used to confirm the position of the mutation site (center: –20.367, 10.904, 9.435; size: 30 × 30 × 30) (*Trott and Olson, 2010*). The ligand molecules CIP were downloaded from PubChem (CID 44469321; https://pubchem.ncbi.nlm.nih.gov/compound/44469321). The ligand was used to dock with Gnina (*Eberhardt et al., 2021*). The affinity score and binding pose were chosen only from the highest convolutional neural network score results from docking simulations. The models were visualized using the PyMOL Molecular Graphic System and Discovery Studio.

## Cross-resistance to TQ and ATO in mutant parasites

The efficacy of TQ and ATO against BgATP4$^{WT}$, BgATP4$^{L921V}$, and BgATP4$^{L921I}$ was evaluated using a fluorescence assay, as described above. The IC$_{50}$ values were calculated from the fluorescence values using non-linear regression analysis (curve fitting) in GraphPad Prism 9 (GraphPad Software Inc, USA).

## Efficacy of CIP plus TQ combination therapy in SCID mice infected with *B. microti*

Twenty 6-week-old female immunocompromised mice (C.B-17/lcrJcl-Prkdcscid strain, CLEA Japan, Inc) were intraperitoneally injected with $1.0 × 10^7$ of *B. microti*-infected blood cells (*Tuvshintulga et al., 2022*). When the average parasitemia across all mice reached 1% (at 4 DPI), the drug treatments were initiated. The CIP group (*n* = 5) was orally administered 20 mg/kg CIP for 7 days. The TQ group (*n* = 5) was orally administered a single dose of 20 mg/kg TQ. The CIP plus TQ group (*n* = 5) received the combination treatment, following the dosage and administration methods described above. The vehicle group (*n* = 5) orally received 0.2 ml of sesame oil as the control. Thin blood smears were prepared every other day and examined under a light microscope to determine parasitemia levels.

## qPCR analysis

To further verify the presence or absence of *B. microti* DNA in treated SCID mice, real-time quantitative PCR analysis was performed as described previously (*Vydyam et al., 2024*). Briefly, genomic DNA was extracted from blood samples (200 µl) collected at 90 DPI and subjected to qPCR analysis using SYBR Green I, targeting a highly conserved region of *Babesia* mitochondrial genome (mtDNA). Primers used were: Bmic-F 5′-TTGCGATAGTAATAGATTTACTGC-3′ and B-lsu-R2 5′-TCTTAACCCAAC TCACGTACCA-3′ (*Qurollo et al., 2017*). The reaction mixture consisted of: 1× Advanced Universal SYBR Green Super Mix (2×) (1725270; Bio-Rad); 0.5 µM of each primer, and 100 ng of genomic DNA.

# Detection of *B. microti* ATP4 gene mutations in relapsed infected SCID mice

The genomic DNA of the mutant parasites was extracted and sequenced (*Ji et al., 2022a*). The primer sets used for sequencing are listed in **Supplementary file 2**. The single-nucleotide variants were confirmed by pairwise alignment to the *Bm*ATP4$^{WT}$ sequence (GenBank: BMR1_03g01005).

## Statistical analysis

Data analysis, namely one-way ANOVA and two-tailed unpaired *t*-tests, was performed using GraphPad Prism (La Jolla, CA, USA) version 9. A p value of <0.05 was considered a statistically significant result.

## Acknowledgements

This work was supported by a Grant-in-Aid for Scientific Research (22H0250906), the JSPS Core-to-Core program, both from the Ministry of Education, Culture, Sports, Science, and Technology of Japan, and a grant from the Strategic International Collaborative Research Project (JPJ008837) promoted by the Ministry of Agriculture, Forestry, and Fisheries of Japan.

## Additional information

### Funding

| Funder | Grant reference number | Author |
| --- | --- | --- |
| Ministry of Education, Culture, Sports, Science and Technology | 22H0250906 | Xuenan Xuan |
| Ministry of Agriculture, Forestry and Fisheries | JPJ008837 | Xuenan Xuan |

The funders had no role in study design, data collection, and interpretation, or the decision to submit the work for publication.

### Author contributions

Hang Li, Data curation, Formal analysis, Investigation, Writing – original draft, Writing – review and editing; Shengwei Ji, Software, Validation, Investigation, Methodology; Nanang R Ariefta, Software, Formal analysis, Methodology; Eloiza May S Galon, Visualization, Writing – review and editing; Shimaa AES El-Sayed, Miako Sakaguchi, Visualization, Methodology; Thom Do, Investigation, Methodology; Lijun Jia, Data curation, Visualization; Masahito Asada, Investigation, Visualization; Yoshifumi Nishikawa, Resources, Methodology; Xin Qin, Data curation, Methodology; Mingming Liu, Conceptualization, Resources, Data curation, Supervision, Funding acquisition, Investigation, Methodology, Writing – review and editing; Xuenan Xuan, Conceptualization, Formal analysis, Supervision, Validation, Methodology, Project administration, Writing – review and editing

### Author ORCIDs

Hang Li http://orcid.org/0009-0008-6361-8955
Shengwei Ji http://orcid.org/0000-0001-7380-4809
Nanang R Ariefta http://orcid.org/0000-0002-3012-6005
Eloiza May S Galon https://orcid.org/0000-0003-2411-8401
Masahito Asada https://orcid.org/0000-0003-0042-6856
Yoshifumi Nishikawa http://orcid.org/0000-0001-5005-6377
Xin Qin http://orcid.org/0000-0001-6448-5180
Mingming Liu http://orcid.org/0000-0002-5935-6736
Xuenan Xuan http://orcid.org/0000-0003-2780-110X

### Ethics

All animal experiments were performed according to the Guide for the Care and Use of Laboratory Animals of Obihiro University of Agriculture and Veterinary Medicine, which was also approved by the Committee on the Ethics of Animal Experiments at the Obihiro University of Agriculture and

Veterinary Medicine, Japan (permit numbers: animal experiment, 22-145 and 23-132; DNA experiment, 2207 and 2208; pathogen, 202308 and 202306).

Reviewer #2 (Public review): https://doi.org/10.7554/eLife.101128.4.sa1
Reviewer #3 (Public review): https://doi.org/10.7554/eLife.101128.4.sa2
Author response https://doi.org/10.7554/eLife.101128.4.sa3

---

# Additional files

## Supplementary files
Supplementary file 1. Primer sets of *B. gibsoni* ATP4.

Supplementary file 2. Primer sets of *B. microti* ATP4.

Supplementary file 3. Interactions of CIP from docking simulations.

MDAR checklist

## Data availability
All data have been deposited in the Dryad Digital Repository and can be accessed via https://doi.org/10.5061/dryad.8kprr4z02.

The following dataset was generated:

| Author(s) | Year | Dataset title | Dataset URL | Database and Identifier |
|---|---|---|---|---|
| Li H, Ji S, Ariefta NR, Galon EMS | 2025 | Efficacy and mechanism of action of cipargamin as an antibabesial drug candidate | https://doi.org/10.5061/dryad.8kprr4z02 | Dryad Digital Repository, 10.5061/dryad.8kprr4z02 |

---

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
