## [Editor Report · eLife Assessment]

This study presents **valuable** findings with practical and theoretical implications for drug discovery, particularly in the context of repurposing cipargamin CIP for the treatment of Babesia spp. The evidence is **solid** with the methods, data, and analyses broadly supporting the claims. The paper will be of great interest to scientists in drug discovery, computational biology, and microbiology.

---

## [Referee Report · Reviewer #2 (Public review)]

Summary:

In this manuscript, the authors present the repurposing of cipargamin (CIP), a known drug against plasmodium and toxoplasma against babesia. They proved the efficacy of CIP on babesia in the nanomolar range. In silico analyses revealed the drug resistance mechanism through a single amino acid mutation at amino acid position 921 on the ATP4 gene of babesia. Overall, the conclusions drawn by the authors are well justified by the data presented. I believe this study opens up a novel therapeutic strategy against babesiosis.

Strengths:

The authors have carried out a comprehensive study. All the experiments performed were carried out methodically and logically.

---

## [Referee Report · Reviewer #3 (Public review)]

Summary:

The authors aim to establish that cipargamin can be used for the treatment of infection caused by Babesia organisms.

Strengths:

The study provides strong evidence that cipargamin is effective against various Babesia species. In vitro growth assays were used to establish that cipargamin is effective against Babesia bovis and Babesia gibsoni. Infection of mice with Babesia microti demonstrated that cipargamin is as effective as the combination of atovaquone plus azithromycin. Cipargamin protected mice from lethal infection with Babesia rodhaini. Mutations that confer resistance to cipargamin were identified in the gene encoding ATP4, a P-type Na ATPase that is found in other apicomplexan parasites, thereby validating ATP4 as the target of cipargamin. A 7-day treatment of cipagarmin, when combined with a single dose of tafenoquine, was sufficient to eradicate Babesia microti in a mouse model of severe babesiosis caused by a lack of adaptive immunity.

Weaknesses:

Cipargamin was tested in vivo at a single dose administered daily for 7 days. Despite the prospect of using cipargamin for the treatment of human babesiosis, there was no attempt to identify the lowest dose of cipagarmin that protects mice from Babesia microti infection.

Comments on revisions:

The authors have edited the manuscript and, in doing so, have addressed all queries pertaining to experimental design. The authors have decided to keep the discussion unchanged, but have replied to this reviewer regarding comments on interpretation of some data. The reader could have benefited from the authors' explanation. Nonetheless, the manuscript in its present form describes a valuable and significant body of work.

---

## [Author Response]

The following is the authors’ response to the previous reviews

**Public Reviews:**

**Reviewer #2 (Public review):**
Summary:In this manuscript, authors have tried to repurpose cipargamin (CIP), a known drug against *Plasmodium* and *Toxoplasma* against *Babesia*. They proved the efficacy of CIP on *Babesia* in nanomolar range. *In silico* analyses revealed the drug resistance mechanism through a single amino acid mutation at amino acid position 921 on the ATP4 gene of *Babesia*. Overall, the conclusions drawn by the authors are well justified by their data. I believe this study opens up a novel therapeutic strategy against babesiosis.Strengths:Authors have carried out a comprehensive study. All the experiments performed were carried out methodically and logically.

We appreciate your positive feedback. Your acknowledgment reinforces our commitment to rigor and thoroughness in our research.

**Reviewer #3 (Public review):**
Summary:The authors aim to establish that cipargamin can be used for the treatment of infection caused by *Babesia* organisms.Strengths:The study provides strong evidence that cipargamin is effective against various *Babesia* species. In vitro growth assays were used to establish that cipargamin is effective against *Babesia bovis* and *Babesia gibsoni*. Infection of mice with *Babesia microti* demonstrated that cipargamin is as effective as the combination of atovaquone plus azithromycin. Cipargamin protected mice from lethal infection with *Babesia rodhaini*. Mutations that confer resistance to cipargamin were identified in the gene encoding ATP4, a P-type Na+ ATPase that is found in other apicomplexan parasites, thereby validating ATP4 as the target of cipargamin. A 7-day treatment of cipagarmin, when combined with a single dose of tafenoquine, was sufficient to eradicate *Babesia microti* in a mouse model of severe babesiosis caused by lack of adaptive immunity.

Thank you for the comments and for your time to review our manuscript.

Weaknesses:Cipargamin was tested in vivo at a single dose administered daily for 7 days. Despite the prospect of using cipargamin for the treatment of human babesiosis, there was no attempt to identify the lowest dose of cipagarmin that protects mice from *Babesia microti* infection. In the SCID mouse model, cipargamin was tested in combination with tafenoquine but not with atovaquone and/or azithromycin, although the latter combination is often used as first-line therapy for human babesiosis caused by *Babesia microti*.

Thank you for your insightful comments. We agree that using a single daily dose over 7 days is one of the limitations in the in vivo trial. Our main goals were to demonstrate cipargamin's efficacy and understand its antibabesial agent mechanism. For future work, we plan to conduct dose‐optimization studies to determine the lowest effective dose in vivo. Regarding the drug combination in the SCID mouse model, although atovaquone and/or azithromycin are frequently used as first-line therapies for human babesiosis, resistance to these traditional drugs is emerging. Based on this challenge, we opted to evaluate a combination with tafenoquine as a novel partner, aiming to overcome resistance issues and improve therapeutic outcomes.

**Recommendations for the authors:**

**Reviewer #2 (Recommendations for the authors):**
None other than some minor grammatical mistakes.

We have corrected the grammatical mistakes.

**Reviewer #3 (Recommendations for the authors):**
The revised manuscript is much improved. I have the following comments.Comment 1: Atovaquone plus azithromycin is effective against *Babesia microti* (Figure 1C) but not against *Babesia rodhaini* (Figure 1E). It would be valuable to provide a possible explanation.

Thank you for highlighting this issue. One potential explanation is that *B. microti* and *B. rodhaini* might have intrinsic differences in drug sensitivity and susceptibility. A previous study reported that both species possess a unique linear monomeric mitochondrial genome with a dual flip-flop inversion system, which generates four distinct genome structures (Hikosaka et al., 2012). In addition, previous studies have shown that mitochondria-associated energy production is greater in *B. microti* than in *B. rodhaini* (Shikano et al., 1998). This suggests that *B. microti*, whose metabolism is largely driven by mitochondrial function, may be more susceptible to drugs (like atovaquone) that induce parasite death by disrupting mitochondrial targets such as cytochrome b (Wormser et al., 2010). Moreover, *B. rodhaini* tends to proliferate more rapidly and causes acute infections, which may outpace any drug effects. Further, the rapid proliferation of apicomplexan parasites, as is the case in *Plasmodium* (Salcedo-Sora et al., 2014), *Theileria* (Metheni et al., 2015), and *B. rodhaini* (Rickard, 1970; Shikano et al., 1995), has been ascribed to glycolysis as the primary energy source. This may have contributed to the reduced efficacy of atovaquone and azithromycin in *B. rodhaini*-infected mice in the current study. Nonetheless, we plan to explore these interspecies differences in our future work.

Comment 2: The relapse that follows a 7-day treatment with cipargamin is transient in BALB/ mice infected with *Babesia rodhaini* (Figure 1E) but persistent in SCID mice infected with *Babesia microti* (Figure 5C). It would be valuable to provide a possible explanation.

Thank you for your insightful comment. One possible explanation is the difference in immune status between the two mouse models. BALB/c mice have a fully functional immune system that can likely clear residual parasites following a transient relapse after cipargamin treatment. In contrast, SCID mice lack an adaptive immune response, which might allow residual *B. microti* parasites to persist and cause a sustained relapse. Additionally, intrinsic differences between *B. rodhaini* and *B. microti*, such as growth rate or drug susceptibility, could also play a role. We plan to explore these factors in future studies.

Comment 3: The effect of cipargamin on parasite pH is the greatest when assessed 4 to 8 min after exposure is initiated (Figure 3E). Yet, resistance of parasites that carry a mutation in ATP4, the target of cipargamin, was assessed 20 min after cipargamin addition. At this time point, cipargamin has very little effect (Figure 3E). Accordingly, data reported in Figure 3G are of limited value.

Thank you for your comment. The initial pH increase we see around 4 to 8 minutes likely reflects the rapid inhibition of ATP4-mediated Na⁺/H⁺ exchange by cipargamin, which quickly alkalinizes the cell. However, after the initial increase, compensatory processes, such as proton influx or metabolic acid production, gradually restored the pH, resulting in a later decline. Although assessing the pH level at 20 minutes may have recorded less dramatic changes, it still allowed us to compare the sustained differences between wild-type and mutant strains. We agree that including earlier time points for the mutants might provide further insight and we will consider this in our future work.

Comment 4: In Figure 3H, please report the lack of statistical significance between wild-type parasites and parasites that carry the mutation L921V.

In Figure 3H, the ATPase activity in erythrocytes infected with wild-type parasites (6.31 ± 1.20 nmol Pi/mg protein/min) is higher than that of the L921V mutation (5.11 ± 0.50 nmol Pi/mg protein/min), but the difference is not statistically significant (*P* = 0.095), so no asterisk was added.

Comment 5: Tafenoquine was administered as a single 20 mg/kg dose. Please specify whether this dose is for tafenoquine succinate or tafenoquine base.

Thank you for raising this point. In our study, the single 20 mg/kg dose refers to tafenoquine succinate. We have clarified this detail in the revised manuscript (Line 40).

Comment 6: A single dose of 20 mg/kg tafenoquine succinate was first tested in the SCID mouse model of severe babesiosis by Mordue et al (JID 2019), not by Liu et al. (JID 2024). Please amend discussion accordingly (line 311). As correctly stated in the discussion, the single 20 mg/kg dose was not sufficient to prevent relapse of *Babesia microti* in the study by Mordue et al. Please provide a possible explanation for why no parasitemia was detected for 90 days in your SCID model (Figure 5C).

Thank you for your comment. We have modified the suggested citation (Line 309). As noted by Mordue et al. (JID 2019), a single 20 mg/kg dose of tafenoquine succinate was insufficient to prevent relapse in their SCID mouse model using *B. microti* (ATCC 30221 Gray strain). In our study, however, no parasitemia was detected for 90 days (Figure 5C) using the *B. microti* Peabody mjr strain (ATCC PRA-99). Differences in the parasite strain and the timing of treatment relative to infection may have contributed to the extended suppression of parasitemia observed in our study. We plan to explore these aspects in future work.

Comment 7: Real-time PCR was used to confirm eradication of *Babesia microti* infection (Figure 5D). Please specify the blood volume from which genomic DNA was extracted for each mouse. Please specify the amount of genomic DNA (i.e., not the volume) used in each reaction. Please explain how/why the cut-off was set at 35 cycles. What were the Ct values when blood was obtained from uninfected mice? For infected mice treated with cipargamin plus tafenoquine, there was no amplification. Was each reaction subjected to a maximum of 40 cycles (as suggested by Figure 5D)?

In our qPCR assay, genomic DNA was extracted from 200 µL of blood per mouse (Line 458). In each reaction, we used 100 ng of genomic DNA (Line 464), and the thermocycling conditions were set at 40 cycles. We set the cut-off at 35 cycles based on our optimization experiments: samples with Ct values ≤ 35 consistently indicated the presence of parasite DNA, while samples without parasite DNA (distilled water and DNA from uninfected mice) had CT values > 35 cycles or undetectable. Although each reaction was run for 40 cycles, for our analysis, we defined samples as negative if no signal was observed beyond cycle 35. In mice treated with cipargamin plus tafenoquine, no signal was detected until 40 cycles, indicating the absence of parasite DNA in the samples.

Comment 8: Persistence of parasite DNA in blood of tafenoquine treated mice highlights the limitation of PCR to assess persistence of infection. That is, PCR cannot distinguish between viable parasites and non-viable (or dead) parasites. An adoptive transfer of blood to immunocompromised mice can help determine whether persistence of DNA is due to persistence of viable parasites. Because the experiment was carried out in SCID mice, no adoptive transfer is needed. Few parasites are required for a successful infection of immunocompromised mice (SCID mice included). Given that parasitemia never rose following treatment of SCID mice with a single dose of tafenoquine, it is highly likely that parasite DNA detected on day 90 post-infection in these tafenoquine treated mice came from persistent non-viable/dead parasites.

We appreciate your comment and acknowledge that the use of PCR has limitations in differentiating between live and dead parasites. It is possible that the residual DNA may represent a small population of dormant parasites that are not actively replicating and thus remain below the detection threshold of parasitemia. Even in highly immunocompromised SCID mice, such dormant parasites might persist without causing overt infection under our experimental conditions. An adoptive transfer experiment in SCID mice, although not strictly necessary, could validate whether the detection of low levels of DNA comes from viable parasites capable of reactivating under different circumstances. Future studies using more sensitive viability assays or adoptive transfer approaches could provide further insights into this possibility.